# The Differential Diagnosis of Congenital Developmental Midline Nasal Masses: Histopathological, Clinical, and Radiological Aspects

**DOI:** 10.3390/diagnostics13172796

**Published:** 2023-08-29

**Authors:** Michal Kotowski

**Affiliations:** Department of Pediatric Otolaryngology, Poznan University of Medical Sciences, 60-572 Poznań, Poland; mkotowski@ump.edu.pl

**Keywords:** nasal dermoid, nasal dermoid sinus cyst, nasal glial heterotopia, nasal glioma, encephalocele, imaging, histopathology, clinical manifestation

## Abstract

Developmental midline nasal masses including nasal dermoids (NDs), encephaloceles (EPHCs), and nasal glial heterotopias (NGHs) are a consequence of disrupted embryonal developmental processes in the frontonasal region. Surgery is the only method of treatment in order to prevent local and intracranial inflammatory complications as well as distant deformities of the facial skeleton. Due to their rarity, similar location, and clinical and radiological symptoms, meticulous preoperative differential diagnostics is mandatory. The aim of this thorough literature review was to present and discuss all clinical, histopathological, and radiological aspects of NDs, NGHs, and EPHCs that are crucial for their differential diagnosis.

## 1. Introduction

A disruption of embryonal developmental processes in the frontonasal region may result in the formation of a group of rare midface malformations described as developmental midline nasal masses. These lesions include nasal dermoids (NDs), encephaloceles (EPHCs), and nasal glial heterotopias (NGHs). Although related to identical locations and considered for many years as disorders of similar embryological origin, many aspects of these disorders are the subject of ongoing debate.

NDs constitute the majority of congenital midline nasal masses, accounting for at least 60% of these abnormalities [1,2]. The term covers both epidermoid cysts and nasal dermoid cysts, sinuses, and fistulas. Several hypotheses concerning the underlying facial developmental processes resulting in a variety of clinical manifestations of NDs have been discussed [3]. None of the theories presented so far sufficiently explains the aberrative processes leading to ND formation [3].

EPHCs are described as a herniation of intracranial components through the developmental defect of the skull base. They are defined depending on their content. Meningoceles are the protrusion of meninges, whereas meningoencephaloceles are masses containing meninges, cerebrospinal fluid (CSF), and brain tissue [4,5,6]. Depending on the anatomic location, EPHCs can be classified as basal or sincipial [7]. A defect in the cribriform plate or the body of the sphenoid resulting in herniation is described as a basal EPHC. According to Adil et al., four subtypes may be distinguished: transethmoidal, sphenoethmoidal, transsphenoidal, and sphenoorbital [4,5,8,9,10]. Only the first two of these manifest as nasal masses [5]. A sincipial EPHC represents the herniation of intracranial material through an osseous and dural defect in the frontoethmoidal region. Therefore, it is also called a frontoethmoidal EPHC. The location of the internal skull defect is in the midline in sincipial EPHCs, but the external skull defect may have various manifestations in the facial bony structure [4,11]. Accordingly, they may be subdivided into nasofrontal, nasoethmoidal, and nasoorbital types [5,12,13]. Mahatumarat et al. proposed the addition of a fourth subtype: combined nasoethmoidal and nasoorbital [14].

NGHs are similar to EPHCs to some degree, but the main difference is that NGHs represent an absence of communication with the CNS or subarachnoid space [15,16]. A mass consists of dysplastic neurglial tissue from the cranial cavity or spinal canal [17,18]. NGH formation is thought to be a consequence of an entrapment of neuroectodermal tissue during the dural closure or a form of disconnected nasal EPHC [15,19]. Nasal glioma, a term previously used to describe these lesions, implies a neoplasm and therefore is a misnomer regarding this congenital developmental abnormality of non-neoplastic origin.

The majority of these developmental masses are located in the nasal region, extranasally or intranasally, or in its immediate proximity. Surgery is the only method of treatment in order to prevent local and intracranial inflammatory complications, as well as distant craniofacial deformities [20]. Due to their rarity, similar location, clinical symptoms, and potential sequelae, meticulous preoperative differential diagnostics is mandatory. Although imaging studies are regarded as fundamental to differential diagnosis, thorough physical examination and histopathological studies also play a crucial role.

The aim of this literature review was to present and discuss in detail the clinical, histopathological, and radiological aspects of NDs, NGHs, and EPHCs that are crucial for differential diagnostics.

## 2. Clinical Manifestations

### 2.1. Nasal Dermoids

NDs may present as a cyst, sinus, or fistula located anywhere from the philtrum of the upper lip to the glabellar region [3]. A hair visible in the skin ostium is considered to be pathognomonic for ND [21] (Figure 1). NDs may be noticeable at birth, presenting as a lump or a pit on the nasal dorsum. The majority of them are diagnosed at infancy or in early childhood, but there are some reports of later diagnoses [22,23,24,25,26,27,28]. A slow-growing mass may be observed in some cases in early infancy [18]. Half of children may present hypertelorism and a broadened nasal bridge. Palpation may reveal a subcutaneous mass in 30% of patients, typically at the level of the osteocartilaginuous junction [1]. Some authors have reported intracranial epidermoid and dermoid cysts in the proximity of the crista galli or foramen caecum without typical nasal manifestation [1]. Intermittent sebaceous discharge from the skin ostium may be observed. Multiple ostia and cysts at more than one level are a rarity [24,25]. The presence of a sinus opening increases the risk of infection and further complications. The probability of infection is estimated to be 7% per year throughout childhood. Owusu-Aim et al. revealed that 50% of children have at least one local infection by the age of 4, and more than 90% before their 9th birthday [29]. Recurring infections influence the long-term results of surgical treatment [30]. Intracranial involvement is estimated in 20% of cases [8,9] but ranges from 4% to 57% in different studies, and it is impossible to determine clinically [21,23,24,27,31,32,33] as there is no correlation between the location of the sinus ostium or cyst and the presence of an intracranial extension [23,27,31]. The consequences of infection include local abscesses, periorbital cellulitis, osteomyelitis, meningitis, cerebral abscesses, or cavernous sinus thrombosis [23,25]. Intranasal masses (intraseptal or nasopharyngeal) are a rarity but may cause nasal obstruction in these cases.

An ND is a non-expansile and non-pulsatile mass that is not compressible and does not transilluminate [34,35]. The mass does not enlarge either during spontaneous crying or as a result of the compression of the jugular veins (negative Furstenberg’s sign) and the Valsalva maneuver [35,36].

### 2.2. Congenital Encephaloceles

EPHCs may manifest as a nasal mass, a widened nasal bridge, hypertelorism, telecanthus, epiphoria, a nasal obstruction, or feeding difficulties depending on the location [5,13,20,37,38]. The physical and endoscopic examination of the nose may reveal a septal deviation, a unilateral tumor typically located superiorly and laterally to the anterior margin of a middle turbinate (Figure 2), or a cerebrospinal fluid (CSF) rhinorrhea [38].

Approximately 15% of all EPHCs are located anteriorly and called sincipial or frontoethmoidal EPHCs. Among them, nasoethmoidal, nasofrontal, and nasoorbital EPHCs can be distinguished. A nasoethmoidal EPHC is diagnosed when the herniation protrudes through the osseous defect between the ethmoid and nasal bone (through the foramen caecum). In these cases, a pathological, bluish mass is located on the nasal root or in the nasal cavity [1]. In nasofrontal EPHCs, the defect is present between the frontal and nasal bones (at the level of the former fonticulus frontalis); therefore, the lesion manifests as a mass at the glabella or lower forehead, anteriorly to the nasal bones [1,5,13]. The medial canthal location of the mass is characteristic for nasoorbital EPHCs [1].

To avoid further facial deformities during growth and potential CNS infectious consequences, all sincipial encephaloceles should be surgically treated at an early age [4].

Basal EPHCs are intranasal, smooth “polypoid”, pink or bluish, and covered with mucous membrane masses [1,5,39]; a nasal obstruction, neonatal respiratory distress, or meningitis in a child may be their first manifestation [1,5,40]. EPHCs are usually pulsatile [18], transilluminating, soft, and compressible during palpation [5].

In contrast to NDs, the Furstenberg sign is positive as EPHCs enlarge (and, additionally, their pulsation becomes more exposed) with jugular vein compression, the Valsalva maneuver, or spontaneously during crying [1,5,36].

EPHCs may be accompanied by other craniofacial abnormalities, such as malformations of cortical development, median cleft face syndrome, and hydrocephalus [13,41]. The morning glory syndrome that is a consequence of the congenital dysplasia of the optic nerve is present in 2/3 of children with a basal EPHC [41]. On the contrary to occipital EPHCs, anterior EPHCs are not connected with neural tube anomalies [1].

Due to the similarities in clinical manifestation, EPHCs have to be taken into consideration in the differential diagnostics of EPHCs, NDs, and NGHs.

### 2.3. Nasal Glial Heterotopias

An NGH is regarded as an EPHC that has lost the intracranial connection [1]. NGHs are usually detected at birth, during infancy, or in early childhood [1,5] but tend to grow along with the patient [1]. There is a male predominance in incidence, but NGHs do not run in families [5]. Depending on the location, these lesions may be divided into extranasal (60%), intranasal (30%), or mixed (combined intra- and extranasal) forms (10%) [1,5,16,18]. Although there is no patent intracranial connection, its remnant in the form of a fibrous stalk to the meninges is present in 15–20% of cases [8,9,20,42].

Extranasal glial heterotopias are usually seen at the glabella or along the nasal dorsum [1]. They may be ocassionally located paramedially, on the side of the nose at the medial canthal region [1,43] (Figure 3). Hipertelorism and a widened nasal bridge are commonly visible, but in some cases only subtle findings such as telecanthus may be revealed [44,45]. The mass is smooth, well-circumscribed, reddish or bluish, and often has telangiectasis on its surface [1]. Such an appearance may lead to a misdiagnosis of capillary hemangioma [42,46]. NGHs are firm and non-compressible during palpation, and they do not transilluminate. An increase in intracranial pressure (ICP) has no influence on these lesions. The Furstenberg sign is negative, and there is no noticeable enlargement during crying [40]. In the study by Penner et al., the diameter of NGHs ranged from 1 to 7 cm (mean 2.4 cm) [16].

Intranasal glial heterotopias are attached to the middle turbinate or lateral nasal wall [5]. These lesions rarely arise from the nasal septum [5,8,9,42,46]. Nevertheless, the development of a mass may result in nasal septum deviation [5]. Intranasal masses are usually “polypoid” and pale in their appearance. They may result in nasal obstruction, mucosal congestion, and epiphora resulting from lacrimal system obstruction [36].

## 3. Histopathological Findings

### 3.1. Epidermoid/Dermoid Lesions

The term dermoid cyst customarily covers a few histopatologically different entities. It has been used to describe epidermoid cysts, true dermoid cysts, and teratoid cysts [47]. Teratoid cysts are composed of ecto-, meso-, and endodermal components [48]. Both epidermoid and dermoid cysts are lined by an ectodermal-derived stratified squamous epithelium [49]. Epidermoid cysts represent an accumulation of desquamated epithelium in the cavity [1]. The lack of dermal appendages, such as sebaceous and sweat glands and hair follicles, distinguishes epidermoid from dermoid cysts. The glands located in the wall of dermoid cysts or sinuses produce sweat and sebum in their lumen. These secretions and their breakdown products consist of a characteristic oily substance including fat and hair [50]. Although clinically identical and hardly distinguishable on imaging, these two lesions are histologically different.

### 3.2. Nasal Glial Heterotopias

Not only are there some clinical similarities between NGHs and EPHCs, but the histopathological differences have also not been clearly determined. Both lesions are composed of varying proportions of neurons and glia [16].

The dysplastic neuroglial components mainly include astrocytes, but occasionally also gemistocytic astrocytes, neurons, or ependymal cells. Although Penner et al. reported ependymal components and neurons in 20% of their samples, usually no significant leptomeningeal, ependymal, or choroid plexus structures are identified [16]. The multinucleated and gemistocystic forms support the recognition of neural tissue [16]. The fibrovascular tissue plays the role of stroma in these masses. Depending on the clinical form and location of NGHs, the external overlying epithelium may present different characteristics. It may present as skin or a metaplastic squamous or typical respiratory epithelium. According to the literature, it may be attenuated or atrophic, but always maintains continuity [16].

Penner at al. revealed that lymphocytes and macrophages or other chronic inflammatory cells were present in all cases of NGH [16]. Moreover, reactive gliosis was routinely observed. There have also been some reports that focal calcifications or cystic structures may be present [51].

The variety of histological pictures requires more than routine hematoxylin-eosin staining. Thus, some additional special staining and immunohistochemistry (IHC) processes have to be implemented [19,52,53,54,55]. Masson trichrome staining is useful, indicating a neural/glial component with a magenta stain and collagen with intense blue. This method may be combined with immunohistochemical reactivity using S-100 protein and glial fibrillary acidic protein (GFAP) antibodies. GFAP is considered to be an astrocyte marker. This allows for highlighting the neural elements over the fibrosis. Alternatively, Nissl staining or neuron-specific enolase may be used to detect neural structures in unclear cases. Reactive fibrosis may be confirmed by immunohistochemical tests for laminin and collagen type IV [16].

It should be highlighted that no significant histopathological differences exist in particular NGHs, regardless of the presence of a central nervous system connection.

Penner et al. identified factors influencing difficulties in the identification of glial components [16]. Intense fibrosis or the accumulation of inflammatory cells makes diagnosis more demanding. The degree of fibrosis may differ, but it is often accompanied by inflammation [16]. Some interesting changes are noticeable according to the age of patients with NGHs. The degree of fibrosis or even sclerosis increases with age [16]. In adult patients, the glial tissue may be hardly identifiable. In these cases, differential diagnosis may be challenging.

### 3.3. Encephaloceles

EPHCs are histologically similar to NGHs. The surface of these pathological masses is covered with a columnar or cuboidal epithelium and dense fibrous dura-like bands [39]. In some cases, the lack of a surface epithelium or dura-like band may be observed. The brain tissue is usually a mature neuroglial tissue composed of neurons and large ovoid or spindle astrocytes. A layer of leptomeninges surrounds these neural structures. Similar to NGHs, chronic inflammatory cells may be present [39]. Barnes claimed that EPHCs in children above the age of 18 months may present a predomination of fibrous tissue over glial cells, with a deficiency of neurons [56].

Mahapatra et al. revealed that none of the frontoethmoidal EPHCs among their group of 92 pediatric cases contained any significant brain elements [57]. The case is different for transphenoidal basal encephaloceles, which often contain important neurological structures, such as the optic pathway, pituitary gland, and hypothalamus [58]. As surgery is aimed at the replacement of critical structures, these forms of EPHCs are extremely demanding to resect.

Standard hematoxylin-eosin staining is supported by immunohistochemistry (S100, vimentin, GFAP). Additionally, brain glial tissue is positive for neuron-specific nuclear protein (NeuN) [39]. NeuN is expressed in the cytoplasm and nuclei of neurons in the postmitotic stage, being characteristic only for mature neuronal cells [59]. To identify the presence of meningeal components, epithelial membrane antigen (EMA) may be implemented [39].

A comparison indicates that NGHs and EPHCs are histologically similar. Some authors claim that cystic structures with ependymal cells are features that distinguish EPHCs from NGHs [60,61]. But others stress that a proper diagnosis is impossible without sufficient data concerning the patient’s preoperative imaging results and intraoperative findings. As biopsies are contraindicated in the case of NGHs and EPHCs, the role of histopathologic examination is confirmative in postsurgical follow-up and may play a role in cases with subtotal resection.

## 4. Imaging Studies

### 4.1. General Considerations

The diagnostic protocol is similar for NDs, NGHs, and EPHCs. Computed tomography (CT) or magnetic resonance imaging (MRI) remain the standards for presurgical management in all midface nasal masses. Regarding NDs, MRI presents slightly higher values of sensitivity, specificity, predictive positive value, and predictive negative value towards intracranial extension in NDs compared to CT [62]. Therefore, MRI plays a pivotal role in diagnostics and should be the first-choice radiological tool. Many researchers agree [1,5]; nevertheless, the complimentary role of CT should be highlighted [63]. Both false-positive and false-negative results are still reported despite advances in radiological imaging. Even regarding solely NDs, numerous needless craniotomies resulting from false-positive imaging findings have been reported [23,25]. The most frequent radiological pitfalls have been described in previous publications [62]. These usually stem from uncompleted endochondral ossification processes of anterior cranial fossa [23,33,63].

As children below the age of 5 are more predisposed to false-positive and false-negative results on imaging regarding intracranial involvement, the combination of CT and MRI should become a gold standard in the diagnosis of nasal dermoids in that age group [62]. Therefore, meticulous radiological studies are fundamental to avoid disastrous consequences in the case of developmental nasal masses.

### 4.2. Nasal Dermoids

CT enables the most effective assessment of crucial bony elements (Figure 4). Some characteristic radiological signs for the intracranial extension of NDs on CT scans have been described in the literature. A widening of the foramen caecum and bifid crista galli remain indirect evidence suggestive of anterior skull base involvement in these cases [23,25,28,35]. Herrington et al. claim that, in addition to the visualization of bony structures, CT also allows for the direct visualization of the transosseous part of the fibrous tract, which may remain invisible in MR [64].

MRI, with its higher resolution regarding soft tissues, reveals direct and indirect signs of intracranial penetration [63,65,66]. NDs represent a variable signal intensity on MRI depending on the protein content [13]. Adil and Rahbar claim that NDs also present circumscribed masses that are hyperintense on T2 images on MRI [67] (Figure 5), although Hedlund stated that dermoids and epidermoids are variable in intensity on T2-weighted imaging [1]. The fatty component is high in NDs, which is why they are hyperintense on T1-weighted images [65,66]. Intravenous enhancement facilitates the delineation of the cyst and sinus tract; differentiation among non-enhancing dermoid cysts, enhancing nasal mucosa, and other enhancing masses (hemagiomas and teratomas); and possible infection [33,68]. If combined with fat-suppressed T1-weighted images, it becomes extremely useful in the differentiation between skull base defects and enhancing non-ossified cartilage of the anterior cranial fossa in the pediatric population [33,64,68].

The diffusion-weighted imaging technique in MRI (DWI) plays an important role in differential diagnosis. NDs are typically represented by high-signal-intensity lesions with a corresponding low signal intensity on apparent diffusion coefficient maps (ADC) [64,68]. Adil and Rahbar pointed out that the decreased diffusivity may facilitate differentiation from low-grade tumors and vascular abnormalities [67]. Both abovementioned tumors and vascular lesions present high signal on DWI ADC maps. High-grade tumors and abscesses similarly present low signal intensity in terms of diffusivity but manifest enhancing soft tissue or surrounding inflammatory changes, respectively. Herrington et al. claim that non-echo-planar diffusion-weighted techniques facilitate the reduction of artifacts from nearby structures [64].

The differentiation between dermoid and epidermoid cysts may be supported by the diffusion-weighted imaging (DWI) technique. This may reveal diffusion restriction, which is considered to be characteristic of epidermoid cysts but may also be observed in dermoid cysts [69,70]. Rodriguez et al. stated that a dermoid cyst should present with the density of fat on CT scans, unlike an epidermoid cyst, which typically presents with the density of water [68].

As dermoid and epidermoid lesions are hardly distinguishable on imaging [13], some authors suggest that the differential diagnosis of dermoid and epidermoid cysts should include a biopsy. This is highly controversial, as both epidermoid and dermoid cysts are treated by complete surgical excision. Moreover, as has been proved in recent studies, any previous incomplete surgical interventions correspond to higher recurrence rates and worse long-term cosmetic results [30].

### 4.3. Encephaloceles

An accurate preoperative diagnosis is of utmost importance in these lesions, as it allows one to prevent catastrophic sequlae [8,9]. The majority of researchers claim that MRI is fundamental in the diagnosis of EPHCs (Figure 6 and Figure 7). It enables one to show the herniation of intracranial tissue and its continuity with the brain. On the contrary, Arifin et al. did not perform routine MRI in their group of 388 pediatric patients and stated that MRI gave no additional critical information to plan a surgical approach [11].

Some authors have highlighted the supportive role of digital angiography or angioresonance in evaluation for the presence of possible vascular structures in EPHCs [71]. Cerebral angiography may be useful in determining potential defects in the main cerebral vessels supplying the sac of EPHCs [72].

Adil et al. highlighted that severe difficulties in proper diagnosis using MRI may appear in the case of a superimposed infection in an older child with a congenital EPHC. It can be hardly distinguishable from sinusitis with a bony defect and secondary EPHC [5].

### 4.4. Nasal Glial Heterotopias

Differential diagnosis focuses on the confirmation of discontinuity with the brain parenchyma [5,13]. The presence of a bony defect on CT scans may be revealed despite the lack of communication with the brain parenchyma [19,63,73]. MRI usually confirms a well-circumscribed, rounded, or polypoid mass that is isointense (rarely hypointense) to gray matter on T1-weighted imaging [1]. The neural tissue in NGHs is in most cases more hyperintense on T2-weighted images compared to normal brain parenchyma [5]. It results from the fact of reactive gliosis and the presence of dysplastic neuroglial components [1,5,74]. (Figure 8).

Dysplastic tissue usually corresponds to no enhancement [1,74]; however, some studies have reported moderate enhancement [75]. Noticeable enhancement at the lesion periphery is a consequence of compression on the surrounding mucosa [1,74]. The effectiveness of MRI in the visualization of a fibrous stalk connection to the CNS is variable [74]. The imaging technique using MRI allows for the assessment of potential nasal or pharyngeal obstruction and accompanying CNS abnormalities [76,77].

The most important clinical, histopathological, and radiological features of NDs, NGHs, and EPHCs are summarized in Table 1.

### 4.5. Role of Ultrasonography

The similarity between hemangioma and NGHs in clinical manifestation and, to some degree, in MRI increases the risk of a potential pitfall in diagnosis [1,74]. Some investigators have proposed the implementation of Doppler ultrasonography to effectively differentiate these two lesions [1,46]. NGHs present a low diastolic flow velocity, whereas hemangiomas present a high diastolic flow velocity in Doppler studies [1,46].

## 5. Future Directions of Research

None of the proposed theories concerning the embryopathological processes underlying the formation of the discussed nasal masses are convincing and unambiguous alone. The recent research has focused on the subject of the molecular and genetic background of these congenital abnormalities. Because of the complexity of the processes involved in the abnormal development of the facial region, many factors are still unknown [3]. The further analysis of the molecular mechanisms, their signaling pathways, and potential genes contributing to the development of the dura mater of anterior cranial fossa and the midface will be crucial in understanding the pathophysiology of NDs, NGHs, and EPHCs [78,79,80,81].

## Figures and Tables

**Figure 1 diagnostics-13-02796-f001:**
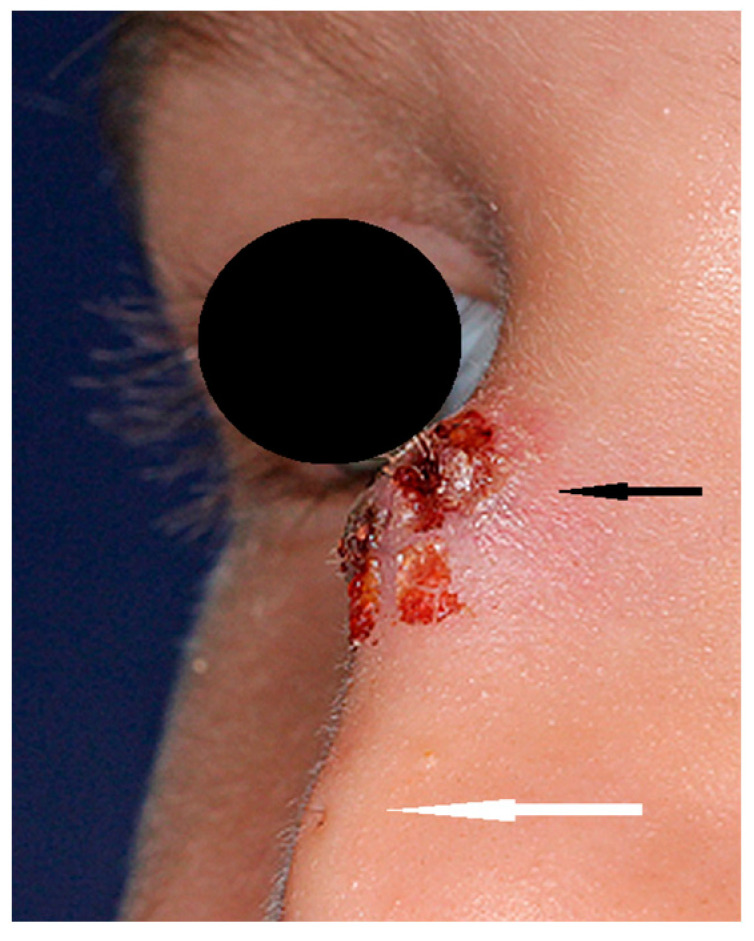
Nasal dermoid (black arrow) with visible skin ostium (white arrow) and hair.

**Figure 2 diagnostics-13-02796-f002:**
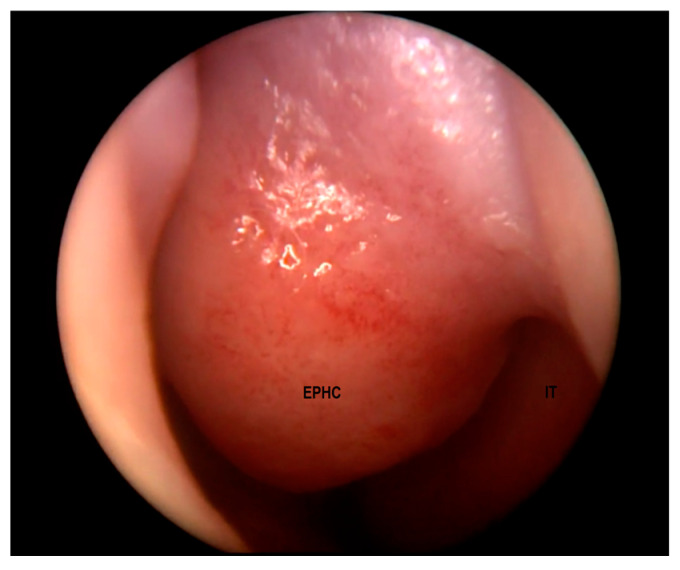
Encephalocele in the left nasal cavity (EPHC). Inferior turbinate (IT) visible laterally.

**Figure 3 diagnostics-13-02796-f003:**
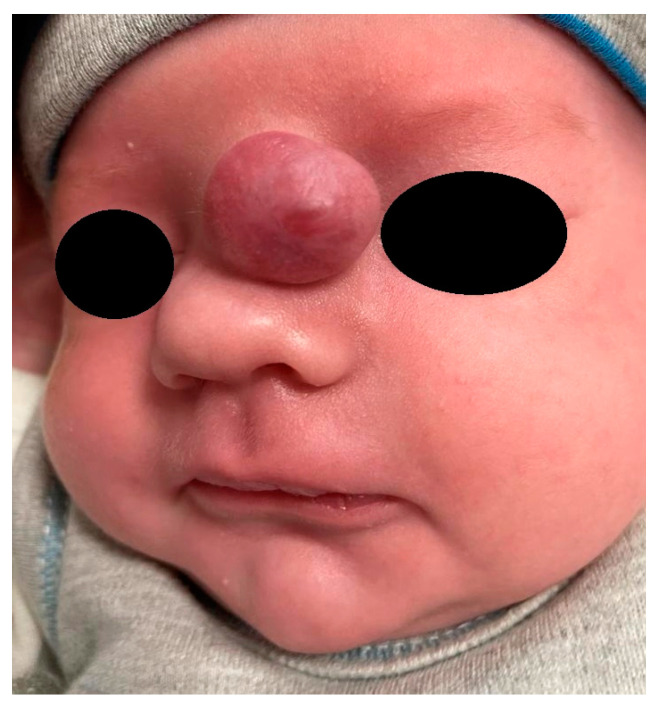
Nasal glial heterotopia (NGH).

**Figure 4 diagnostics-13-02796-f004:**
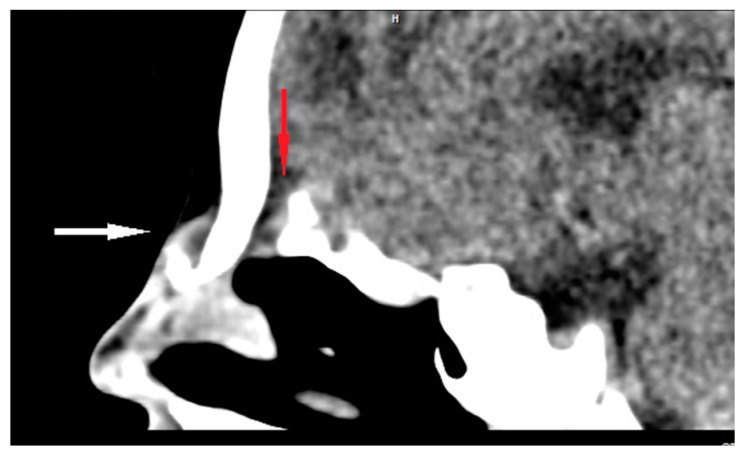
Sagittal CT scan of ND (white arrow) with intracranial extension (bony defect anteriorly to the crista galli—red arrow).

**Figure 5 diagnostics-13-02796-f005:**
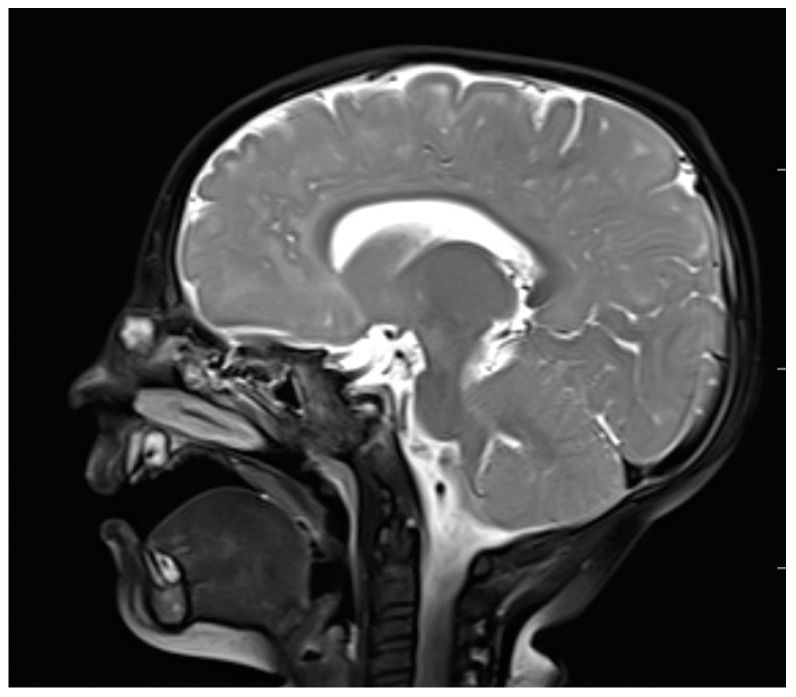
Nasal dermoid—MRI–T2 (sagittal view).

**Figure 6 diagnostics-13-02796-f006:**
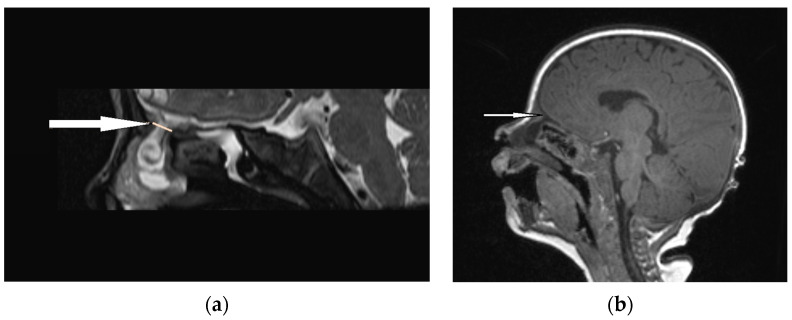
EPHC (sagittal view): (**a**) MRI–T2—a herniation of intracranial tissue (white arrow); (**b**) MRI–T1—a communication with subarachnoid space (white arrow).

**Figure 7 diagnostics-13-02796-f007:**
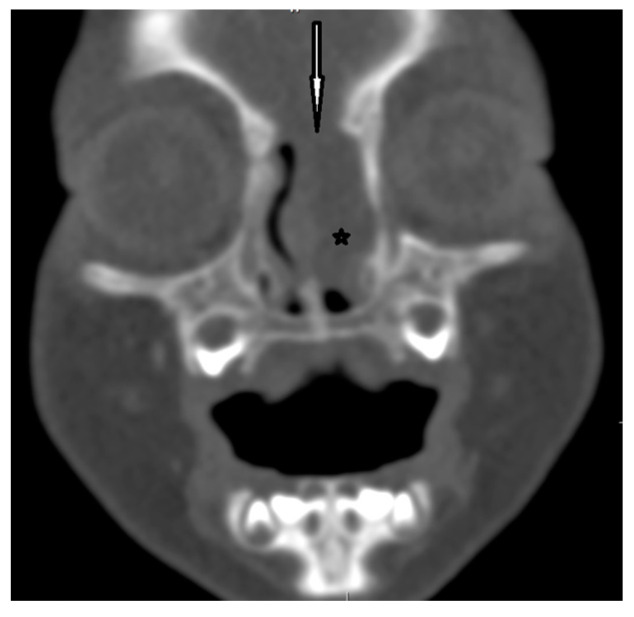
EPHC–CT—coronal view (the white arrow indicates the bony defect; asterisk—intranasal mass of EPHC).

**Figure 8 diagnostics-13-02796-f008:**
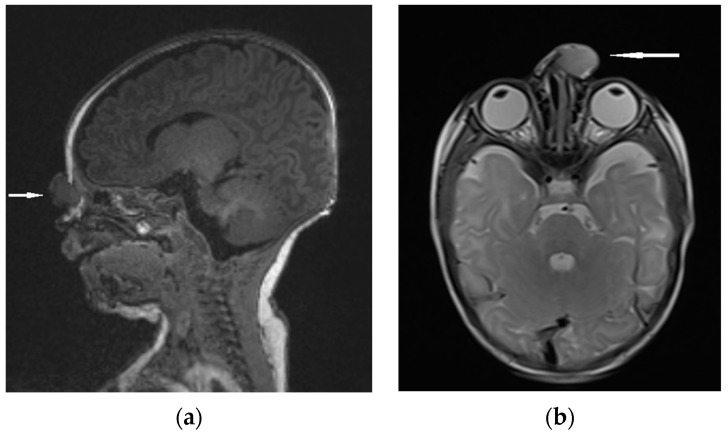
NGH—the extranasal mass indicated by white arrows: (**a**) MRI sagittal view; (**b**) MRI–T2 axial view.

**Table 1 diagnostics-13-02796-t001:** The summary of the most important clinical, radiological, and histopathological features of NDs, NGHs, and EPHCs.

		Type of Pathology		
		ND	NGH	EPHC
Clinical manifestation	Location	Anywhere from the philtrum of the upper lip to the glabellar region	Usually glabella, nasal dorsum, or intranasal	Transethmoidal, sphenoethmoidal, transsphenoidal, sphenoorbital, nasofrontal, nasoethmoidal, or nasoorbital
	Form	Cyst, sinus, or fistula; possible intracranial extension	Extranasal, intranasal, or combined mass	Intra- or extranasal mass
	Characteristics	Hair visible in the skin ostium; non-expansile, non-pulsatile, non-compressible, and non-transilluminating mass	Tends to grow along with the patient; firm and non-compressible mass, non-transilluminating; intranasal—pale, “polypoid”; extranasal—smooth, well-circumscribed, reddish or bluish, often with telangiectasis on its surface	Smooth “polypoid” mass, pink or bluish, covered with mucous membrane, usually pulsatile, transilluminating, soft, and compressible
	Furstengerg’s sign	Negative	Negative	Positive
Radiological characteristics	CT	No correlation between the particular location of the sinus ostium or cyst and the presence of intracranial extension; bifid crista galli and widening of foramen caecum (suggestive of intracranial extension); dermoid cyst—density of fat; epidermoid cyst—density of water	Bony defect may be revelaed	Developmental bony defect of the skull base
	MRI	Variable signal intensity depending on the protein content; fat-suppressed T1-weighted images—differentiation between skull base defects and enhancing non-ossified cartilage of anterior cranial fossa; DWI—typically high-signal-intensity lesion with corresponding low signal intensity on ADC maps	Discontinuity with the brain parenchyma; variable visualization of a fibrous stalk connection to CNS; well-circumscribed, rounded, or polypoid mass—isointense or rarely hypointense to gray matter on T1-weighted imaging; neural tissue—more hyperintense on T2-weighted images to normal brain parenchyma in most cases; dysplastic tissue usually corresponds with no enhancement or moderate enhancement; noticeable enhancement at the lesion periphery	Herniation of intracranial tissue and its continuity with the brain
Histopathological findings		Lined by ectodermal-derived stratified squamous epithelium; consists of a characteristic oily substance including fat and hair	Varying proportions of neurons and glia; dysplastic neuroglial components including mainly astrocytes, but occasionally also gemistocytic astrocytes, neurons, or ependymal cells; usually no significant leptomeningeal, ependymal, or choroid plexus structures; reactive gliosis—routinely observed; degree of fibrosis/sclerosis increasing with age; external overlying epithelium represents different characteristics (depending on NGH clinical form and location)	Histologically similar to NGHs; covered with columnar or cuboidal epithelium and dense fibrous dura-like band; layer of leptomeninges surrounding neural structures; brain tissue—usually a mature neuroglial tissue composed of neurons and large ovoid or spindle astrocytes
	Staining/IHC		Hematoxylin-eosin staining; Masson trichrome staining; Nissl staining or neuron-specific enolase (neural structures in unclear cases); immunohistochemistry— S-100, GFAP (astrocytes marker); laminin and collagen type IV—for reactive fibrosis	Hematoxylin-eosin staining; immunohistochemistry—S100, vimentin, GFAP; glial tissue—positive to NeuN; EMA to identify the presence of meningeal components

## Data Availability

Not applicable.

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
