# Peer review of "The Differential Diagnosis of Congenital Developmental Midline Nasal Masses: Histopathological, Clinical, and Radiological Aspects"

_diagnostics, 2023, doi:10.3390/diagnostics13172796_

Round 1
Reviewer 1 Report
This narrative review on congenital middle line nasal masses is interesting. To my opinion only minor english language correction is required. I think that the addition of a table summarizing their statements would be of use
no further comments
Author Response
General information: I would like to thank the Reviewer for the revision.
Reviewer’s comment: This narrative review on congenital middle line nasal masses is interesting. To my opinion only minor english language correction is required.
Reply: I appreciate your assessment of the manuscript. The manuscript was checked by the native speaker and some language corrections have been made.
Reviewer’s comment: I think that the addition of a table summarizing their statements would be of use.
Reply and action: Thank you for this valuable comment. The table has been added.
Reviewer 2 Report
In his review, the author tried to underline the clinical, histopathological and radiological aspects of nasal dermoid, congenital encephalocele and nasal glial heterotopias and what should be considered in the differential diagnosis.
The article as a whole is very nicely arranged, well written and very well organized. I can contribute to the article at this point. I think it would be better if the author enriches the article by presenting examples of his own cases with radiological and endoscopic images of nasal dermoid, congenital encephalocele and nasal glial heterotopias.
Kind regards
Author Response
General information: I would like to thank the Reviewer for the revision.
Reviewer’s comment: The article as a whole is very nicely arranged, well written and very well organized. I can contribute to the article at this point.
Reply: I appreciate your assessment of the manuscript.
Reviewer’s comment: I think it would be better if the author enriches the article by presenting examples of his own cases with radiological and endoscopic images of nasal dermoid, congenital encephalocele and nasal glial heterotopias.
Reply and action: Thank you for this comment. I have added the images representing discussed pathologies to the manuscript.